SciPost Physics

Submission

# Similarity between the kinematic viscosity of quark-gluon plasma and liquids at the viscosity minimum

Matteo Baggioli[1] & Kostya Trachenko[2] & Vadim Brazhkin[3]

**1** Instituto de Fisica Teorica UAM/CSIC, c/Nicolas Cabrera 13-15, Universidad Autonoma de Madrid, Cantoblanco, 28049 Madrid, Spain.
**2** School of Physics and Astronomy, Queen Mary University of London, Mile End Road, London, E1 4NS, UK
**3** Institute for High Pressure Physics, RAS, 108840, Troitsk, Moscow, Russia
* Corresponding author: mbaggioli@ifae.es

December 24, 2020

## Abstract

Recently, it has been found that the kinematic viscosity of liquids at the minimum, $\nu_m$, can be expressed in terms of fundamental physical constants, giving $\nu_m$ on the order of $10^{-7}$ m$^2$/s. Here, we show that the kinematic viscosity of quark-gluon plasma (QGP) has a strikingly similar value and support this finding by experimental data and theoretical estimations. We discuss the implications of this result for understanding the QGP including the similarity of its shear flow to that in liquids at the viscosity minimum. The similarity may also indicate that the QGP is close to the dynamical crossover. Our results provide quantitative evidence for the universality of the shear diffusivity proposed earlier.

## 1  Introduction

The quark-gluon plasma (QGP) is a state of matter emerging above the deconfinement QCD transition at $T \approx 1.8 \times 10^{12}$ K [1]. It is produced by highly energetic collisions [2–4] and can be thought of as a plasma made of quarks and gluons. The inter-particle interactions in QGP are strong and can not be treated using conventional theoretical methods such as perturbation theory. Flow and viscosity are the properties of QGP which have probably been discussed most [5–12]. More recently, perturbative QCD and data-driven phenomenological approaches [13–18] were used to describe the viscosity of QGP. One indication of this analysis is that viscosity of QGP is temperature-dependent, and that

this dependence is important to describe the experimental data.

In condensed matter physics, predicting liquid viscosity from theory and without modelling has not been possible for the same reason related to strong interactions. Liquid viscosity strongly depends on temperature and pressure. Viscosity is additionally strongly system-dependent and is governed by the activation energy barrier for molecular rearrangements, $U$, which in turn is related to the inter-molecular interactions and structure. This relationship in complicated in general, and no universal way to predict $U$ and viscosity from first principles exists. This is appreciated outside the realm of condensed matter physics [19]. Tractable theoretical models describe the dilute gas limit of fluids where perturbation theory applies, but not dense liquids of interest here [20]. The same problem of strong interactions or, phrased differently, the absence of a small parameter, were believed to disallow a possibility of calculating liquid thermodynamic properties in general form [21]. For example, the theoretical calculation and understanding of the liquid energy and heat capacity has remained a long-standing problem [22] which started to lift only recently when new understanding of phonons in liquids came in [23].

Despite these complications, there is one particular regime of liquid dynamics where viscosity can be calculated in general form and, moreover, expressed in terms of fundamental physical constants. We have recently found [24] that the kinematic viscosity *at its minimum*, $\nu_m$, is

$$\nu_m = \frac{1}{4\pi} \frac{\hbar}{\sqrt{m_e m}} \tag{1}$$

where $m_e$ and $m$ are electron and molecule masses.
The kinematic viscosity $\nu$ is equivalent to the diffusion constant of the shear diffusion mode $D_s$ (transverse momentum diffusivity), and we will be referring to these properties interchangibly in this paper depending on the context. Eq. (1) follows from writing viscosity at the minimum in terms of two UV cutoff parameters, the interatomic separation and Debye vibration period, and subsequently using fundamental relations such as Bohr radius and Rydberg energy setting these UV parameters in condensed matter phases. For atomic hydrogen where $m$ is given by the proton mass $m_p$, (1) results in the fundamental minimal kinematic viscosity as

$$\nu_m = \frac{1}{4\pi} \frac{\hbar}{\sqrt{m_e m_p}} \approx 10^{-7} \ \frac{\mathrm{m}^2}{\mathrm{s}} \tag{2}$$

We show the experimental kinematic viscosity for several liquids from Ref. [24] in Fig. 1. The experimental minima in Fig. 1, $\nu_m^{exp}$, agree with Eq. (1) by a factor 0.5-3 for different liquids [24]. $\nu_m^{exp}$ are in the range of about

$$\nu_m^{exp} = (0.5 - 2) \cdot 10^{-7} \ \frac{\mathrm{m}^2}{\mathrm{s}} \tag{3}$$

in agreement with (2).

Here, we show that the kinematic viscosity of the QGP has a value strikingly similar to $\nu$ in liquids at the minimum and close to $10^{-7} \ \frac{\mathrm{m}^2}{\mathrm{s}}$ as in (2) and (3). We use experimental data as well as theoretical estimations to back up this result. We discuss the implications of the similarity between liquids and QGP from the point of view of shear flow and particle dynamics. We also propose that the similarity may indicate that the QGP is close to the *dynamical crossover*. Finally, our results provide quantitative evidence for the universality of the shear diffusivity proposed earlier.

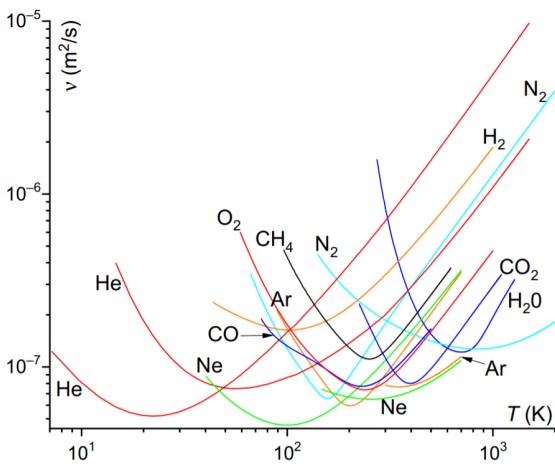

Figure 1: Experimental kinematic viscosity $\nu = \frac{\eta}{\rho}$ of noble, molecular and network liquids showing minima around $\nu = 10^{-7} \frac{m^2}{s}$. $\nu$ for $H_2$, $H_2O$ and $CH_4$ are shown for pressure $P = 50$ MPa, 100 MPa and 20 MPa, respectively. $\nu$ for He, Ne, Ar and $N_2$ are shown at two pressures each: 20 and 100 MPa for He, 50 and 300 MPa for Ne, 20 and 100 MPa for Ar and 10 and 500 MPa for $N_2$. The minimum at higher pressure is above the minimum at lower pressure for each fluid. The experimental data of $\eta$ and $\rho$ are from Ref. [25].

## 2  Kinematic viscosity

We make two preliminary observations which will become useful for the subsequent discussion. First, we recall the dynamics of particles at the minimum of the liquid viscosity where Eq.(1) applies. This minimum is due to the dynamical crossover [23, 26, 27] separating (a) the liquid-like dynamics combining oscillatory and diffusive components of molecular motion where viscosity decreases with temperature and (b) purely diffusive gas-like motion where the viscosity increases with temperature. The crossover between these two regimes implies that viscosity has a minimum. These minima are experimentally seen in liquids, as illustrated in Fig.1. At the crossover where molecules lose the oscillatory component of motion and continuously move diffusively over distances comparable to an interatomic separation, the viscosity can be evaluated by assuming that the particle mean free path $L$ is approximately equal to the inter-particle distance $a$. This results in Eq.(1), in agreement with the experimental viscosity minima [24]. This mechanism will become useful in the discussion below. We note that viscosity minimum also appears in a different mechanism involving relativistic hydrodynamics and considering the effects of short-wavelength hydrodynamic fluctuations [28]. The minimum in this picture is related to the breakdown of the hydrodynamics expansion [29]. A minimum for $\eta/s$ was also discussed in holographic models where it corresponds to a transition between thermal gas background and a big black hole solution at high temperature [30], and in certain nuclear matter models [31, 32].

Second, we observe that the minimal viscosity in liquids (1) is consistent with the uncertainty relation. As discussed above, the viscosity at the minimum corresponds to $L = a$ and can be written as $\eta = \rho v a = \frac{m}{a^3} v a = \frac{pa}{a^3}$, where $p$ is the particle momentum and $\rho = \frac{m}{a^3}$ is density. Combining this with the uncertainty relation $pa \geq \hbar$ gives $\eta \geq \frac{\hbar}{a^3}$, or

$$\nu_m \geq \frac{\hbar}{m} \qquad (4)$$

| $E/V$ | 1 GeV/fm$^3$ [33] |
|---|---|
| $\eta$ | $5 \cdot 10^{11}$ Pa· s [8] |
| $m_p$ | $1.67 \cdot 10^{-27}$ kg |
| $a_p$ | $0.84 \cdot 10^{-15}$ m |
| $a$ | $0.5 \cdot 10^{-15}$ m [35] |
| $T_{\mathrm{QGP}}$ | $2 \cdot 10^{12}$ K [8] |

Table 1: Parameters used to estimate the properties of QGP.

Therefore, the uncertainty relation gives a weaker bound on $\nu$ as compared to $\nu_m$ in (1): $\frac{\hbar}{m}$ in (4) is smaller than $\nu_m$ in (1) by a factor $\frac{1}{4\pi}\left(\frac{m}{m_e}\right)^{\frac{1}{2}}$, which is in the range 5–23 for liquids shown in Fig. 2. For atomic hydrogen, Eq.(4) gives the lower bound as

$$\nu_m \geq \frac{\hbar}{m_p} \approx 10^{-7} \ \frac{\mathrm{m}^2}{\mathrm{s}} \tag{5}$$

which is close to the experimental minima in (3). The bound (5) is smaller than (2) by the factor $\frac{1}{4\pi}\left(\frac{m_p}{m_e}\right)^{\frac{1}{2}} \approx 3$.

We now calculate the experimental value of kinematic viscosity of QGP. In a system with conserved number of particles, $\nu = \frac{\eta}{\rho}$, where $\eta$ is dynamic viscosity and $\rho$ is density. The density can be estimated in several ways. In a non-relativistic system with conserved particle number, $\rho = \frac{1}{c^2}\frac{E}{V}$, where $\frac{E}{V}$ is energy density (see Table (1)). This gives $\rho \approx 5 \cdot 10^{18}$ kg/m$^3$. We note the earlier discussion that the QGP energy density is not far from the energy density inside nucleons [33] and so the proton density can be used in the order-of-magnitude estimation (all our evaluations corresponds to order-of-magnitude evaluations) as $\rho = \frac{m_p}{a_p^3}$, where $m_p$ and $a_p$ are the proton mass and size. Using the values in Table 1, this estimate gives $\rho \approx 3 \cdot 10^{18}$ kg/m$^3$, close to the previous estimation. However, the QGP is a relativistic charged fluid described by relativistic hydrodynamics [34], where the diffusion constant of the shear diffusion mode $D_s$ (transverse momentum diffusivity), is given by:

$$\nu \equiv D_s = \frac{\eta}{\chi_{\pi\pi}}, \quad \chi_{\pi\pi} = \epsilon + p \tag{6}$$

where $\chi_{\pi\pi}$ is momentum susceptibility given in terms of the energy density $\epsilon$, $p$ is pressure and we set $c = 1$.

Eq. (6), which can be formally derived using relativistic hydrodynamics [34], substitutes the non-relativistic expression $\nu = \eta/\rho$. $\nu$ in relativistic and non-relativistic case are identical under the replacement $\rho \to \epsilon + p$. The momentum susceptibility can be written by using the thermodynamic identity:

$$\epsilon + p = sT + \mu q \tag{7}$$

where $\mu$ and $q$ are chemical potential and charge density, respectively. In the case of QGP, $\mu$ and $q$ are the baryonic chemical potential $\mu_B$ and baryon number density $B$, respectively. In the part of the QCD phase diagram where the QGP is experimentally observed, the baryonic chemical potential is small, $\mu/T \ll 1$, and the second term $\mu q$ can be neglected, leading to:

$$\chi_{\pi\pi} \approx sT \tag{8}$$

This can be seen as follows. The typical energy density of QGP is about $\epsilon \approx 1 \,\mathrm{GeV/fm}^3 = 1.6 \times 10^{35}$ Pa. Assuming an approximate conformal equation of state, we have $\epsilon + p = \frac{4}{3}\epsilon \approx 2.1 \times 10^{35}$ Pa. We can compare this value to the r.h.s. of (7), where $\mu$ is the baryon chemical potential. Taking the QGP temperature as $T \approx 2 \times 10^{12}$ K and using the Kovtun-Son-Starinets (KSS) bound $\eta/s = \frac{1}{4\pi}\frac{\hbar}{k_B}$ [11, 12] (this bound also holds in the presence of finite charge density [36]), we obtain an estimate $sT \approx 1.8 \times 10^{35}$ Pa. This implies that the charge corrections are small and $sT \gg \mu q$. We note that the full knowledge of thermodynamic parameters $(T, \mu_B, \dots)$ can improve the estimation of $\nu_{\mathrm{QGP}}$. Combining (6) and (8) gives

$$\nu_{\mathrm{QGP}} \approx \frac{\eta}{s\,T} \tag{9}$$

Finally, $\frac{\eta}{s}$ can be evaluated using the experimental value for $\eta/s$ as [9]:

$$\left.\frac{\eta}{s}\right|_{\mathrm{QGP}} \approx \frac{3}{4\,\pi}\frac{\hbar}{k_{\mathrm{B}}} = 3\left.\frac{\eta}{s}\right|_{\mathrm{KSS}} \tag{10}$$

where $\left.\frac{\eta}{s}\right|_{\mathrm{KSS}}$ is the KSS bound [11, 12]. This gives

$$\nu_{\mathrm{QGP}} \approx \frac{3}{4\pi}\frac{\hbar c^2}{k_{\mathrm{B}}T} \tag{11}$$

where we restored $c$.

We note that Eq. (11) contains the Planckian relaxation time $\tau_{\mathrm{Pl}} = \frac{\hbar}{k_B T}$ which we will discuss later in the paper. Using the temperature of QGP from Table 1 gives the experimental value of kinematic viscosity of QGP, $\nu_{\mathrm{QGP}}^{exp}$, as

$$\nu_{\mathrm{QGP}}^{exp} \approx 10^{-7}\frac{\mathrm{m}^2}{\mathrm{s}} \tag{12}$$

as for liquids at the viscosity minimum in (2) and (3).
Given that the dynamic viscosity $\eta$ and the density of QGP are about 16 orders of magnitude larger than those values in liquids, the similarity of $\nu$ is striking.

The similarity between the kinematic viscosity of liquids at the minimum and QGP viscosity is further illustrated in Fig.2 where we show $\nu$ for a subset of liquids from Fig.1 for clarity and plot $\nu$ as a function of temperature normalised by a characteristic temperature scale. We plot liquid $\nu$ as a function of $\frac{T}{T_c}$, where $T_c$ is the temperature of the critical point and $\nu_{\mathrm{QGP}}^{exp} \approx 10^{-7}\frac{\mathrm{m}^2}{\mathrm{s}}$ from Eq. (12) as a function $\frac{T_{\mathrm{QGP}}}{T_{cr}}$, where $T_{\mathrm{QGP}}$ is given in Table 1 and $T_{cr} = 1.82 \times 10^{12}$ K is the temperature of the QCD chiral crossover [1]. As before, we observe the proximity of $\nu_{\mathrm{QGP}}^{exp}$ to the minimum of $\nu$ in liquids.
We note that in a system with conserved particle number (a non-relativistic fluid), the viscosity $\eta$ in Table 1 and $\nu_{\mathrm{QGP}}^{exp}$ in (12) correspond to an equivalent density of about $\rho \approx 6.5 \cdot 10^{18}$ kg/m$^3$ and close to our previous density estimations.

Theoretically, we make two observations about $\nu$ of QGP. First, we note that experimental value of kinematic viscosity, $\nu_{\mathrm{QGP}}^{exp} \approx 10^{-7}\frac{\mathrm{m}^2}{\mathrm{s}}$ in (12) is close to $\frac{\hbar}{m_p}$ in (5). Recall that (5) was derived for the liquid hydrogen system. In this system, particles setting viscosity are hydrogen atoms whose interaction is due to electromagnetic forces and whose size and interatomic separation are orders of magnitudes larger than those in the system of protons or partons. However, $\nu$ is insensitive to these details in one particular state of

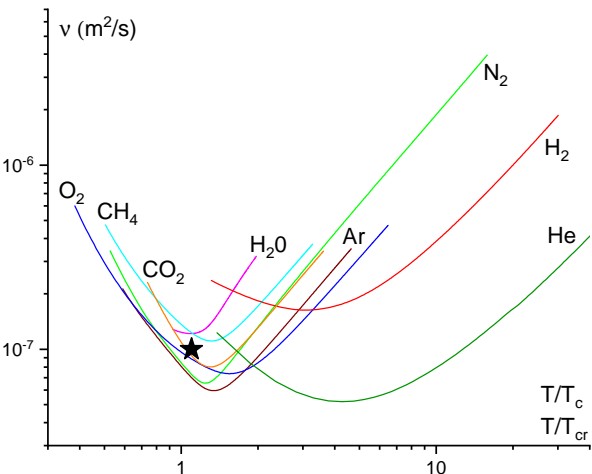

Figure 2: Experimental kinematic viscosity $\nu = \frac{\eta}{\rho}$ of noble, molecular and network liquids plotted as a function of $T/T_c$, where $T_c$ is the critical temperature [25]. $\nu$ are shown at the following pressures: 20 MPa (Ar), 100 MPa ($H_2O$), 10 MPa ($N_2$), 30 MPa ($O_2$), 20 MPa ($CH_4$), 50 MPa ($H_2$), 20 MPa (He) and 30 MPa ($CO_2$). The star shows $\nu_{\text{QGP}}^{exp}$ from Eq. (12) at temperature $T_{\text{QGP}}/T_{cr}$, where $T_{\text{QGP}}$ is in Table 1 and $T_{cr}$ is the temperature of the QCD chiral crossover $T_{cr} = 1.82 \times 10^{12}$ K [1].

the system. This state corresponds to the dynamical crossover where the mean free path $L$ is comparable to the shortest inter-particle spacing $a$. As discussed earlier and elaborated on in more detail below, this regime corresponds to a very special regime of dynamics and to the dynamical crossover at the Frenkel line [23, 26, 27] where liquid viscosity is close to its minimum. We write viscosity at the crossover, $\eta_c$, as $\eta_c \approx \rho v a$, where $v$ is the average velocity of particles and $a$ is inter-particle distance. Using $\rho = m/a^3$ gives $\eta = \frac{p}{a^2}$, where $p$ is particle momentum. Estimating $p$ as $p \approx \hbar/a$ from the uncertainty relation gives $\eta = \frac{\hbar}{a^3}$ and $\nu$ at the crossover, $\nu_c$, as

$$\nu_c \approx \frac{\hbar}{m} \tag{13}$$

which depends on the particle mass only. Setting $m = m_p$ gives $\nu_c \approx 10^{-7} \frac{\text{m}^2}{\text{s}}$ as in (12).

Second, the same result for $\nu$ of about $10^{-7} \frac{\text{m}^2}{\text{s}}$ follows from the estimation of the diffusion constant that features in the mean-square displacement of particle motion, $D$. We note that $D$ is generally different from $D_s$ in Eq. (6). $D$ coincides with $\nu$ and $D_s$ in the gas-like regime of particle dynamics at high temperature where the same momentum transfer mechanism governs both Navier-Stokes and diffusion equations [37] ($D$ and $\nu$ are different in the liquid-like regime at low temperature, where viscosity is $\propto 1/D$). At low temperature where particle dynamics combines oscillatory motion and diffusive jumps between quasi-equilibrium positions and where relaxation time $\tau$ is the time between these jumps, $D \approx \frac{a^2}{\tau}$ [37] (we note that $\tau$ bears no relation to the Israel-Stewart relaxation time $\tau_\pi$ appearing in higher-order relativistic hydrodynamics to overcome the well-known causality problems of the first-order linearised formulation.)As temperature increases and $\tau$ becomes comparable to the shortest timescale in the system $\tau_0$ (in liquids $\tau_0$ is on the

order of Debye vibration period of about 0.1 ps), the oscillatory component of particle motion is lost and particles start moving continuously, corresponding to the dynamical crossover discussed above and the Frenkel line [23, 26, 27]. At the crossover, $D \approx \frac{a^2}{\tau_0}$. The same result follows for the kinematic viscosity $\nu_c = va$ in the regime $L = a$ at the dynamical crossover if $v$ at the crossover is written as $v \approx \frac{a}{\tau_0}$. Approximating the inter-parton distance $a = 0.5$ fm [35] by $a_p$ gives $D = \nu_c \approx \frac{a_p^2}{\tau_0}$. If we relate the shortest timescale $\tau_0$ to the Planckian relaxation time [38]:

$$\tau_{\mathrm{Pl}} = \frac{\hbar}{k_B T} \tag{14}$$

we find

$$D = \nu_c = \frac{a_p^2}{\hbar} \, k_{\mathrm{B}} \, T_{\mathrm{QGP}} \tag{15}$$

The timescale $\tau_{\mathrm{Pl}}$ was related to several fundamental physical phenomena, including the linear resistivity of strange metals [39], universal bounds on quantum chaos [40], bounds on diffusion [41–43], SYK model [44], magic bilayer graphene [45], black holes [46] and holography [47]. $\tau_{\mathrm{Pl}}$ can also appear in transport properties using semi-classical microscopic physical arguments.[1] This is different to the AdS-CFT approach where this timescale emerges from emergent IR criticality [48] and which contrasts experimental results [49] of Planckian transport in high-temperature thermal conductivity where phonons behave classically.

Using the temperature of QGP (see Table 1) gives $D$ in (15) as $D \approx 10^{-7} \mathrm{m}^2/\mathrm{s}$ as in (12). A very small relaxation time $\tau_{\mathrm{Pl}} = \frac{\hbar}{k_{\mathrm{B}} T_{\mathrm{QGP}}} \approx 0.4 \cdot 10^{-23}$ s interestingly contrasts with large experimental viscosity of QGP, $\eta_{\mathrm{QGP}} = 5 \cdot 10^{11}$ Pa·s, which is close to liquid viscosity at the liquid-glass transition. In liquids, this viscosity corresponds to liquid relaxation time $\tau_l = 50 - 500$ s, as follows from using the Maxwell relation $\eta = G\tau$ and a typical high-frequency liquid shear modulus $G = 1 - 10$ GPa. $\tau_l$ is close to that of the solid glass around the liquid-glass transition and is about 15-16 orders of magnitude larger than the shortest time scale of the system given by the Debye vibration period $\tau_0$ on the order of 0.1 ps. Applying the Maxwell relation to QGP, $\eta_{\mathrm{QGP}} = G\tau_{\mathrm{Pl}}$, gives $G \approx 10^{35}$ Pa. Combining it with $G = \chi_{\pi\pi} v^2$, where we neglected pressure in the relativistic case as before, and using $\chi_{\pi\pi}$ from above gives the transverse speed of sound $v$ close to the speed of light. Hence, in condensed matter terms, the QGP is an ultra-dense matter with relativistic speed of excitations but familiar kinematic viscosity close to that in liquids at the minimum and dynamic viscosity close to the system at the liquid-glass transition.

## 3   Discussion

We now discuss the implications of these results. We first note that kinematic viscosity of liquids at the minimum in Eqs. (1) and (13) does not contain the electron charge setting the energy of electromagnetic interactions (the charge cancels out, as does the inter-particle separation) and depends on the particle mass only [24]. This implies that $\nu_m$ applies to systems with different types of inter-particle interactions and distances. In this sense, it is interesting that Eq. (13) with $m = m_p$ gives the same result for the liquid hydrogen at the minimum and for the QGP. This intriguingly implies that useful insights into dynamical and transport properties of the QGP can be gained using the concepts

---

[1]We thank Jan Zaanen for pointing this out.

from condensed matter systems such as ordinary liquids, despite different interactions and different fundamental theory governing the QGP.

Second, the similarity of the kinematic viscosity of liquids at the minimum and $\nu_{\text{QGP}}^{exp}$ implies similar flow dynamics. In the non-relativistic case, the shear velocity field is governed by Navier-Stokes equation $\rho \frac{Dv}{Dt} = \eta \nabla^2 v$, which depends on $\nu = \frac{\eta}{\rho}$ only. $\nu$ also features in the Reynolds number, which governs the dynamical similarity of flows. In the relativistic hydrodynamics relevant to the QGP, the dynamics of shear modes comes from the conservation of the stress energy tensor [34]: $\partial_\mu T^{\mu\nu} = 0$, resulting in the diffusive motion for the shear modes as $\omega_s = -iD_s k^2$, with the difference that in the relativistic case $D_s = \frac{\eta}{\epsilon + p}$ rather than $D_s = \frac{\eta}{\rho}$. As discussed above, the two diffusion constants are approximately similar in the range of QGP parameters. This points to the universality of $D_s$ in terms of its applicability to both relativistic and non-relativistic systems as discussed earlier [41]. We will revisit this point below.

Third, in deriving (13) and (15) we assumed that the mean free path is about the same as the inter-particle distance or $\tau \approx \tau_0$. From the condensed matter perspective, this corresponds to a particular regime of particle dynamics of liquids and a particular line on the phase diagram as discussed above. In this regime, the system is outside the low-temperature regime where particles have a combined oscillatory and diffusive motion and where viscosity *decreases* with temperature and vary by 16 orders of magnitude (relaxation time varies between $10^3$ s at the glass transition and about 0.1 ps at high temperature). The system is also outside the gas-like regime where the mean free part exceeds inter-particle separation and viscosity *increases* with temperature and becomes infinite in the ideal-gas limit [8]. Instead, the liquid is finely tuned to be in between these two regimes and at the *crossover* between the liquid-like and gas-like motion where the oscillatory motion is just lost and the particle mean free path is comparable to the inter-atomic separation and where viscosity has minima (see Fig. 2). The crossover corresponds to the Frenkel line on the phase diagram separating gas-like and liquid-like states of liquids and supercritical fluids [23, 26, 27]. We note that although viscosity minima in Fig.1 and 2 is due to the crossover between the liquid-like and gas-like motion, the temperature and pressure of the viscosity minimum may depend on the path taken on the phase diagram [23]).

In view of current interest and experiments to ascertain the critical point of QGP as well as supercritical behavior of QGP, it will be interesting to explore to what extent the dynamical crossover at the Frenkel line applies to the QGP phase diagram. Having made the assumption that the system is at the dynamical crossover, we found that (a) the calculated $\nu$ of QGP is close to the experimental value of $\nu_{\text{QGP}}^{exp}$ in (12) and (b) these values are close to both experimental and theoretical kinematic viscosity in liquids at the minimum $\nu_m$. This similarity gives an insight into the dynamics of QGP at experimental conditions. The analogy with liquids, if appropriate to pursue further, would indicate that the currently measured QGP is interestingly close to the *dynamical crossover* between the liquid and gas-like states. The analogy with liquids would also indicate that future experiments at higher energy may lift the system from the dynamical crossover into the gas-like regime, corresponding to the increase of fluid viscosity in Fig.2, and will detect a viscous response consistent with gas-like dynamics. In fluids, this regime starts to the right from the minima in Fig.2. We note that sufficiently close to the minima, the system is dense, strongly-interacting and non-perturbative.

Fourth and finally, it was earlier suggested [41] that $\nu$, or transverse momentum dif-

fusivity $D_s$, is a universal property in a sense that it applies to both relativistic and non-relativistic cases, generalizing the previous discussion of relativistic bounds [11] used to discuss the properties of QGP and other systems. The quantitative similarity of $\nu_m$ of two vastly different systems found here supports this view.

# Acknowledgements

We thank K. Behnia, A. Buchel, S. Cremonini, S. Hartnoll, K. Landsteiner, P. Romatschke, K. Schalm and J. Zaanen for fruitful discussions and interesting comments. K. T. thanks EPSRC for support. M. B. acknowledges the support of the Spanish MINECO's "Centro de Excelencia Severo Ochoa" Programme under grant SEV-2012-0249.

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
