# Peer review of "Similarity between the kinematic viscosity of quark-gluon plasma and liquids at the viscosity minimum"

_SciPost Physics_

## Round 1 · Referee Report · Anonymous · 2021-1-31

Strengths

The analysis is timely and directly relevant to the search for the QCD critical point and the more detailed upcoming analysis of the temperature dependence of the viscosity.

Weaknesses

The main weakness of the paper is the lack of a discussion of how the adopted assumptions and simplifications affect the final result (see report for details).

Report

Baggioli, Trachenko and Brazhkin examine the value of the kinematic viscosity of the quark gluon plasma (QGP) and compare it to the kinematic viscosity of liquids, building on previous work on the latter by Trachenko and Brazhkin. In particular, they argue that their estimate of the QGP viscosity is comparable to the minimum value of the viscosity of liquids, roughly on the order of $10^{-7} m^2/s$. This is an interesting result, especially if one could make a strong argument for its origin and robustness, beyond the set of assumptions made in this particular paper.

To arrive at the estimated QGP viscosity the authors make a number of approximations, most notably neglecting the chemical potential and any temperature dependence of $\eta/s$, the shear viscosity to entropy ratio. They choose the latter to take the universal KSS value $1/4\pi$, while generically the quantity is expected to be sensitive to temperature, as the authors themselves point out. While this is a valid first step, it would benefit the manuscript if the authors would comment more extensively on the potential role of temperature dependent effects, and how these may alter their main result. Similarly, what is the expectation once a finite chemical potential is taken into account? As a follow up paper, it may be interesting to examine these points in detail, for a more realistic comparison to the QGP.

As a second point, the authors discuss the role of the uncertainty relation and throughout the analysis rely on a particle-like description of the QGP. Thus, they use elements of kinetic theory and assume the existence of quasi-particles, which may not be present in the strongly interacting QGP. I would like the authors to comment on when such assumptions fail, and how this may affect their arguments and main results.

Once these minor additions are implemented, I will gladly recommend the manuscript for publication. The questions tackled in this work are timely and important, given the efforts in the QCD community to search for the critical point and to better understand the temperature dependence of the viscosity.

Requested changes

1. Discuss the conquequences of dropping the assumption of negligible chemical potential and $\eta/s \sim 1/4\pi$.
2. State more clearly when the authors are assuming kinetic theory, the existence of quasi-particles, whether these are valid assumptions and if not, how they would modify their arguments and result.
These points can be addressed in a qualitative way, since a more quantitative analysis would be more appropriate for follow-up work.

  • validity: -
  • significance: -
  • originality: -
  • clarity: -
  • formatting: -
  • grammar: -

Author:  Matteo Baggioli  on 2021-03-22  [id 1320]

(in reply to Report 1 on 2021-01-31)

reply attached

Attachment:

QGP_1_LuAbda6.pdf

---

## Round 1 · Referee Report · Anonymous · 2021-3-2

Strengths

1. Interesting numerical observation about the fact that many liquids as well as the QGP appear to have kinematical viscosity of similar magnitude.
2. Several observations about why 10^(-7) m^2/s could be a fundamental result.
3. Some interesting discussion as to why this result indicates that QGP is close to crossover.

Weaknesses

1. In equation 2 why aren't the authors using a dressed mass for the electron?
2. I found the usage of \geq in places like eq 4 and 5 grossly misleading. For instance in defining rho the authors drop implicit factors. Surely the \geq should be an approximate statement where there is a O(1) factor that cannot be determined. This makes the discussion around eq 5 questionable in my opinion.
3. In the discussion the authors make an interesting observation that the electric charge cancels out as does the interparticle separation and refer the reader to [24]. I went through [24] and did not appreciate the logic used to arrive at this result. For starters, the way that the cancellation of the electric charge happens appears to involve a ratio of two quantities (below eq 9 in [24]). Since either of these quantities could involve a dressing factor, there could easily be a residual factor arising due to some screening effect. Why is this necessarily O(1)?
4. Is there a direct holographic calculation of kinematic viscosity? To clarify, not using known results for eta,s as in eq 9 but via a direct calculation of response.

Report

While the premise of this paper is interesting, compared to ref [24], the new result here is to extend the observations in [24] to QGP. From my understanding, the reason this is nontrivial is essentially the usage of relativistic fluid dynamics equations for instance eqs 7-10. This makes the current paper quite a bit different than [24]. As discussed in my report, some of the inequalities are stronger than can be justified, while some of the discussion can be improved significantly. I feel after these changes and clarifications are made, the paper may meet the standards of SciPost.

Requested changes

1. Inequalities should be properly justified and limitations clearly discussed. Various implicit O(1) factors should be pointed out.
2. The main point of this paper is to use the result of kinematical viscosity and argue why this suggests that QGP as observed in present experiments are close to crossover. I feel that this needs a separate section and more justification than what is currently provided.
3. Point number 3 in my report above should be addressed.

  • validity: good
  • significance: ok
  • originality: ok
  • clarity: ok
  • formatting: reasonable
  • grammar: good

Author:  Matteo Baggioli  on 2021-03-22  [id 1319]

(in reply to Report 2 on 2021-03-02)

Reply attached.

Attachment:

QGP_1.pdf

---

## Editorial Decision

resubmitted